# Generation of a Specific Fluorescence In Situ Hybridization Test for the Detection of Ovarian Carcinoma Cells

**DOI:** 10.3390/biomedicines12061171

**Published:** 2024-05-24

**Authors:** Amelie Limburg, Xueqian Qian, Bernice Brechtefeld, Nina Hedemann, Inken Flörkemeier, Christoph Rogmans, Leticia Oliveira-Ferrer, Nicolai Maass, Norbert Arnold, Dirk O. Bauerschlag, Jörg Paul Weimer

**Affiliations:** 1Department of Gynaecology and Obstetrics, Christian-Albrechts-University and University Medical Center Schleswig-Holstein, 24105 Kiel, Germany; limburg.amelie@gmx.de (A.L.); nicolai.maass@uksh.de (N.M.); norbert.arnold@uksh.de (N.A.); dirk.bauerschlag@uksh.de (D.O.B.); 2Department of Gynaecology, University Hospital Hamburg-Eppendorf, 20246 Hamburg, Germany

**Keywords:** ovarian cancer, FISH, aCGH, chromoanagenesis

## Abstract

Examinations of ovarian cancer cells require the ability to identify tumor cells. Array-based comparative genome hybridization (aCGH) on 30 ovarian carcinomas (OC) identified three genomic loci (8q24.23; 17p12; 18q22.3) over- or under-represented in OC. A fluorescence in situ hybridization (FISH) probe of these three loci is intended to identify tumor cells by their signal pattern deviating from a diploid pattern. Human DNA from these three loci is isolated from bacterial artificial chromosomes (BAC), amplified and labeled with fluorescent dyes. After a standard FISH procedure, 71 OC suspensions from primary tumors, three OC cell lines, three lymphocyte suspensions, and one mesenchymal cell line LP-3 are analyzed with a fluorescence microscope. On average, 15% of the lymphocytes deviate from the expected diploid signal pattern, giving a cut-off of 36%. If this value is exceeded, tumor cells are detected. The mesenchymal cell line LP-3 shows only 21% as a negative control. The OC cell lines as positive controls exceed this value at 38%, 67%, and 54%. Of the 71 OC primary cultures, four cases fell below this cut-off as false negatives. In the two-sample t-test, the percentages of conspicuous signal patterns differ significantly.

## 1. Introduction

Ovarian cancer (OC) is one of the diagnoses with the worst prognosis in the range of gynecological–oncological diseases. The relative 5-year survival rate is about 40%. A key determining factor is the time of initial diagnosis; 76% of carcinomas are detected in late stage III/IV. The 5-year survival rate when detected in stage I is significantly higher at 88%, and in stage II, it is 79%. Hereditary mutations in the breast cancer genes *BRCA1* and *BRCA2* are the cause in one in ten cases, and if there is a known familial history, regular screening can facilitate earlier diagnosis [1,2]. There are three possible tissue origins of the OC: surface epithelium of the ovary, epithelium of the tube, and mesothelium covering the ovary and forming abdominal cavities [3]. Ninety percent arise from the surface epithelium of the ovary and represent epithelial ovarian carcinoma (EOC) [4]. The different types are divided into Type I and Type II [5]. Type I predominantly describes well-differentiated low-grade neoplasms, whereas Type II describes high-grade neoplasms that arise in the so-called de novo development without known precursor lesions. Type I includes low-grade serious, low-grade endometrioid, clear cell, mucinous, and transitional Brenner tumors. They do not show any *TP53* mutations and are characterized by their genetically relatively stable and indolent character [6]. Type II, on the other hand, mostly describes highly aggressive, poorly differentiated, and highly invasive carcinomas, which have mutations in the *TP53* gene in over 80% of cases. These are usually recognized at a late stage due to their increased formation of ascites, so understanding metastasis via this route is particularly important [5]. The colonization of OC cells in the abdominal cavity does not occur via an endothelial barrier, as is the case with most other solid tumors. In the case of OC, malignant ascites occur in more than a third of initial diagnoses and in every form of recurrent OC [7]. This malignant form plays an essential role in both metastasis and various forms of chemoresistance; the detailed and presumably interconnected processes are part of current research [8]. Primary and secondary tumor foci consist of complex cell types. In addition to tumor cells, there are also connective tissue cells, endothelial cells, and cells of the immune system in a tumor focus [9]. In order to increase an understanding of the complex processes of how these cells interact with each other, it is necessary to be able to differentiate the tumor cells from the other cell types. Because solid tumor cells, in contrast to non-tumor cells, accumulate genomic abnormalities, especially in the OC, genomic markers such as typical copy number variation (CNV) are ideal for distinguishing them from other cells [10,11]. As a result of this chromoanagenesis, typical CNVs can unmask tumor cells if it is known at which locations in the tumor genome chromoanagenesis particularly frequently produces changes. Such information can be explored using array-based comparative genome hybridization (aCGH). Particularly common CNVs can then be detected using a probe for fluorescence in situ hybridization (FISH) with corresponding DNA homology to the respective CNV. We deliberately do not detect tumor-relevant genes here, but rather want to make the genomic chaos in tumor cells visible by the simpler technique of FISH. Here, fluorescently labeled DNA, which corresponds to a specific locus in the genome, is melted into a single strand. Depending on the homology of their base pairing, these molecules hybridize to matching base sequences of the cell DNA, which is also denatured to form a single strand. FISH only indicates the presence of specific loci of the genome under study. Chromosomal changes in the genome of solid tumor cells lead to genomic imbalances. These are more likely to accumulate in some areas of the tumor genome than others. Loci in these more frequently altered areas can serve as markers for tumor cells and can be displayed using FISH. Such detection of tumor cells has already been shown on endometrial carcinoma cells [12]. The aim of this work is to find areas with the most common chromosomal changes in ovarian cancers and then generate a FISH probe homologous to these areas. This OC-FISH test is then intended to detect OC cells in that the hybridization pattern differs from the typical diploid pattern of non-neoplastic cells. In this study, 30 OCs were tested for genomic imbalances by aCGH. The three most common loci were used to generate homologous FISH probes. The validity of the OC-FISH test produced in this way was first checked on ovarian carcinoma cells, lymphocytes and the mesenchymal cell line LP-3. The strength of the OC-FISH test was then validated on 71 cell culture suspensions from primary ovarian carcinoma cell cultures.

## 2. Materials and Methods

Biomaterial: genomic imbalances in 30 ovarian cancers from patients treated at the Clinic for Gynecology and Obstetrics in Kiel between 1996 and 2013 were recorded using aCGH.

As a negative control for the OC-FISH test, we used normal male metaphases and interphases on CGH target slides (Abbot Molecular Inc., Des Plaines, IL, USA), a mesenchymal cell line LP-3, and specially isolated lymphocytes. As a positive control for OC-FISH test, we used three primary ovarian cancer cultures and ovarian cancer cell lines OVCAR 3, OVCAR 8, and SKOV 3. The authenticity of the cell lines was tested and confirmed using STR marker testing. The OC-FISH test was validated on 71 fixed primary cell cultures of ovarian cancer from 1999 to 2010 (Table 1). The use of these patient materials is based on their consent and a vote of the Ethics Committee of the Medical Faculty of the Christian-Albrechts University in Kiel (B327/10).

aCGH: following the Agilent protocol, 1 μg of isolated DNA from a tumor and an equivalent amount of reference Agilent female DNA (Agilent, Santa Clara, CA, USA) were used for aCGH. The DNA was digested with the restriction enzymes ALU I and RSA I. The tumor DNA was then labeled with Cy5-dUTP and the reference DNA with Cy3-dUTP (Agilent, Santa Clara, CA, USA). The labeled samples were purified according to the Agilent protocol and hybridized overnight. The hybridized and washed microarrays (Agilent 2x400k+SNP, design: 028081) were read out using a Dx Microarray Scanner G5761 from Agilent (Agilent, Santa Clara, CA, USA). The raw data were processed with the Agilent Feature Extraction software (Vers.: 3.0.5.1, Agilent Technologies Inc., Santa Clara, CA, USA) and then evaluated with the Agilent Cytogenomics (Vers.: 3.0.6.6) software (Agilent Technologies Inc., Santa Clara, CA, USA).

Preparation of the OC-FISH test: matching genomic sequences that most frequently exhibited CNV in OC were obtained from BACPAC Genomics, Inc. (BPG)—BACPAC Re-source Center; (BACPAC Genomics, Emeryville, CA, USA) bacterial artificial chromosomes (BAC) were used that had a homologous DNA sequence to the most common CNVs. These DH10B *Escherichia coli* clones were cultured overnight with 12.5 μg/mL chloramphenicol at 37 °C according to BACPAC instructions. The BAC DNA was isolated using a Qiagen^®^ Plasmid Mini Kit (Qiagen, Hilden, Germany). As in Weimer et al. (1999), the BAC DNA is labeled with fluorescent dyes using a DOP-PCR [13]. The BAC clones RP11-17M8, RP11-347C14, RP11-449D3, RP11-911D14, and RP11-122H7 are homologous to 8q24.33 and are labeled with Green-dUTP (Enzo Life Science, NY, USA). The BAC clones RP11-455M13, RP11-1133K3, and RP11-504H5 are homologous to 18q22.3 and are labeled with aminoallyl-dUTP Cy3 (Jena Bioscience, Jena, Germany). The BAC clones RP11-837P21, RP11-746E8, and RP11-714I2 are homologous to 17p12 and are labeled with both fluorophores. The labeled DOP-PCR products are mixed with 40 µg Cot-1 DNA (GeneOn-BioScience, Ludwigshafen, Germany) and precipitated in 70% ethanol overnight. The precipitate is centrifuged at 4 °C, 13,000 RPM for 30 min. The dried pellet is dissolved in 50 µL hybridization mix (2×SSC, 50% formamide, 10% dextran sulphate, 1% Tween 20, and pH 7.0) [14,15].

OC-FISH test: the slides were defatted in ethanol (ETOH)/1% HCl for several days before hybridization. After the fixed cell suspensions were applied to the glass surface and buffered in Centipur phosphate buffer (Merck, Darmstadt, Germany) for 4 min, the cells were pretreated for hybridization by incubating the slides for 2 min at 37 °C in 2×SSC/0.5% Igepal, then incubated for 15 min in the 0.005% pepsin solution at 37 °C. The slides were then washed in DPBS for 3 min and fixed in 1% buffered formaldehyde solution for 10 min at room temperature. After a further 3 min in DPBS, the slide was dehydrated in an ascending alcohol series consisting of 70%, 90%, and 100% ETOH for 1 min each and then air-dried at room temperature. Both the target DNA in the prepared cells and the DNA of the OC-FISH test were co-denatured after covering with 11 µL OC-FISH test/slide half at 75 °C. Hybridization takes place overnight at 37 °C. After hybridization, coverslips were removed, and the slide was washed in 2×SSC, 0.1% Igepal for 2 min. This is followed by a stringent washing step in 0.4×SSC, 0.3% Igepal at 72 °C for 2 min. After another wash step in 2×SSC, 0.1% Igepal for 1 min, the slides are dehydrated in an ascending alcohol series (ethanol 70, 90, and 100%) and air-dried. Finally, the slides are capped with mounting medium containing DAPI. The observation was carried out on a Zeiss Axioplan 2 fluorescence microscope (Zeiss, Oberkochen, Germany), which is equipped with optical filter sets for FITC and Cy3. The documentation and evaluation are carried out using the ISIS software (Vers. 5.8, Metasystems, Altlusheim, Germany). One hundred and fifty interphases per case are counted in three series. Each recording to be evaluated is created as a seven-layer Z-stack so that no signals are lost in the three-dimensional core. Signal patterns that deviate from the typical diploid structure of two signals per loci are considered conspicuous. The signals to be considered for the evaluation must be within the core marked by DAPI. Signals are evaluated separately if they have a distance of at least one signal diameter between two signals.

Prospective primary cell culture: ovarian carcinoma material of four prospective patients (UF-478; UF-358; UF-384; UF-004) were cultured immediately after surgery as described by Kurbacher et al. (2011) [16]. Proliferation tests were carried out on the primary cultures, and 3D cell structures were cultured as described by Hedemann et al. (2021) [17].

## 3. Results

The evaluation of the aCGH data on 30 ovarian carcinomas revealed gains and losses that occur repeatedly in up to 60% of all cases at typical genomic loci (Figure 1).

Particularly notable here are gains in chromosome 8q23 (60%), losses in 17p12 (57%), and losses in 18q22.3 in about half of all OCs examined with aCGH. These three genomic locations with the most common CNVs in ovarian cancer serve as a sequence template for a three-color OC-FISH test (Figure 2).

In order to determine a detection limit with which the proportion of deviations from the diploid state by the OC-FISH test can reliably detect ovarian cancer cells, a total of 91 lymphocyte nuclei were counted in three series. In 14 lymphocyte nuclei, the signal constellations deviated from the expected diploid pattern with two signals for each locus. In the three series, the non-diploid signal patterns had a mean of 15.38% and a standard deviation of 6.99%. Three times the standard deviation is added to the mean value found, resulting in the cut-off value of 36% for the OC-FISH test. If the count shows a proportion of non-diploid nuclei above the cut-off value, the existence of non-diploid cells is assumed to be given. This evaluation of the OC-FISH test was initially tested on the diploid mesenchymal cell line LP-3 as a negative control and on ovarian carcinoma cell lines as a positive control. The negative control showed an average proportion of non-diploid cells of 21% in three series, whereas the positive control had 53% nuclei with a deviation from the diploid signal pattern. To validate the OC-FISH test, it was carried out on fixed cells from 71 primary ovarian carcinoma cultures (see Table 1); 67 fixed primary cultures were above the cut-off and 4 below (Figure 3). The two-sample t-test shows a significant difference (t = 0.016) in the proportions of OC cells with an abnormal FISH pattern and cell suspensions that are not tumor cells.

Considering the analysis of the lymphocytes, the negative control, the positive controls, and the validation, there are six true-negative findings, four false-negative findings, 73 true-positive findings, and no false-positive findings. Within this artificial case composition, which does not correspond to the natural population, we achieve a sensitivity of 0.94 and a specificity of 1 with the OC-FISH test. The average sensitivity value of the OC-FISH test for all possible specificity values is 0.84 (AUC), as Figure 4 shows.

In the OC-FISH test, all four prospective primary cultures produced proportional FISH signal patterns that deviated from the diploid pattern and were above the determined cut-off. The proportion of non-diploid signal patterns in UF-478 is 45.76%, in UF-358 is 68.42%, in UF-384 is 48.95%, and in UF-004 is 36.59%. We can therefore assume the existence of tumor cells in all primary cell cultures. Proliferation tests and 3D cell cultures confirm the existence of tumor cells in these prospective primary cultures.

## 4. Discussion

Identifying tumor cells with individual genetic or specific surface markers is only possible to a limited extent [18,19]. The dynamic genetic changes in solid tumor cells also change these specific markers without the cell losing its tumor character. Román-Lladó et al. (2023) and Chrzanowska et al. (2020) have successfully shown that FISH is suitable for detecting tumor cells in different entities [20,21]. Unlike in this study, the authors used tumor-specific driver mutations as a target for the FISH probes. To use FISH as a far-reaching biomarker for tumor cells, the target of the FISH probe should concern tumor-specific DNA loci, which represent the diverse genomic changes in the form of genomic gains and losses. The loci that, on average, show the most deviations from a diploid state like CNVs in a tumor entity are suitable for this purpose.

The analysis of the most common CNVs in ovarian cancer partly corresponds to the data in the literature. Kim et al. (2007) have already shown genomic increases in 8q24 in serous OC, although these were located more distally in a gene-rich region in 8q24.3 [22]. Although our analysis also shows strong gains in this area, our peak of 60% is in 8q24.2 (Figure 1). In nearly 60% of the OCs we examined, we can also confirm losses in the chromosomal region 17p12, as also reported by Kim et al. (2007) and Lambros et al. (2005) [22,23]. The losses in 18q22 that we detected are also confirmed by Lambros et al. and Tan et al. [23,24]. Tan et al. (2011) confirm comparable gains and losses in clear cell ovarian carcinomas even in all three loci [24]. Despite different types of OC, the CGH results of these three authors largely correspond to our data, although we deliberately included all OC types in our study. The concordance of the CNVs suggests that the process of chromoanagenesis in OC results in comparable gains and losses [10,11]. Under this condition, a genomic marker that indicates the loci with the most frequent changes should be suitable for detecting OC cells. We were able to generate such a genomic marker calling OC-FISH test. To produce the OC-FISH test, we used the BACPAC genome bank, which offers *E-coli* clones with BACs. We ordered partially overlapping or at least adjacent clones and used their BACs to produce the OC-FISH test (Figure 2). During production, it turned out that two things are advantageous. On the one hand, staining three loci with only two colors (by staining two loci with only one color but staining the third loci evenly with both colors) has the advantage that the selection of colors with little background leads to clean and clear results. The more different colors are used, the higher the probability that one of the colors used will make the entire coloring unusable due to its own background. Furthermore, it is essential that the interphases to be analyzed are captured in their 3D structure by a focused Z-stack. This is the only way to ensure that all signals from the OC-FISH test are recorded. The signal length of the individual loci extends over a clearly visible 500 to 770 kb (Figure 2). Even if the genes *FBXO15*, *TIMM21*, and *CYB5A* are present in the loci 18q22 and the genes *MYOCD* and *ARHGAP44* in the loci 17p12, the selection of these loci for the OC-FISH test is not about these genes but primarily about the high probability that a CNV may be present in the three loci, which can be detected with the OC-FISH test. Other FISH tests in gynecology and senology are more about detecting the amplification of relevant genes, such as human epidermal growth factor receptor 2 (HER2/neu) [25]. This OC-FISH test is intended to detect typical genome changes, such as chromoanagenesis, as a biomarker proof. The basic functionality of this test is documented by hybridization on an inconspicuous metaphase shown in Figure 2. Here, it is also clear that the signals are detected at the intended location, and there are no cross-hybridizations. In principle, deviations from the diploid pattern are also possible in diploid, intact cells due to spatial overlays of signals. This level of misinterpretation must first be determined so that the existence of OC cells can actually be assumed in the quantity. The counting of signal constellations that deviated from the diploid pattern resulted in an average of 15.38%. The standard deviation was 6.99%. We added these three times to the average and thus arrived at a limit of 36%. The existence of true non-diploid cells can be assumed if a signal distribution that deviates from the diploid pattern occurs more frequently than in 36% of the interphases examined. We tested whether this regulation can also be used for cells other than lymphocytes using the mesothelial cell line LP-3 as a negative control and on a primary cell line of ovarian carcinoma as a positive control. Both LP3 and the primary OC cell line could be clearly differentiated using the OC-FISH test (Figure 3). The validation on 71 fixed primary cultures of ovarian cancer also showed very good differentiation by the OC-FISH test, with only four false-negative results, all of which were above the average for LP-3 and lymphocytes, compared to 67 true-positive results. At this point, it is important to realize that the retrospective compilation of the tested OC patients does not correspond to the natural prevalence but was artificially concentrated. An attempt to clarify the actual cell type of the four false-negative OC cell lines using immunohistochemical markers failed because the fixed cell suspensions were all over 10 years old, and the antibody epitopes were apparently too degenerated. The receiver operating characteristic curve (ROC) reflects an area under the ROC curve of 0.8434 and, thus, represents an acceptable quality of the OC-FISH test. The calculated sensitivity of the OC-FISH test is 0.94 (Figure 4).

For future studies of expression data and interactions in tumors, it will be important to determine the proportion of tumor cells and diploid cells in the microenvironment. This information will be particularly important when developing 3D cell culture models. The OC-FISH test can be helpful in an experimental approach for better and more targeted interpretation of the test results of further investigations. The OC-FISH test has the advantage over the FISH tests mentioned above in that it does not indicate specific gene mutations that are associated with tumor development. These can be very different and diverse. Rather, the OC-FISH test indicates very frequently occurring CNVs, which, unlike gene-specific mutations, can be detected relatively frequently.

## 5. Conclusions

With the OC-FISH test, we now have an instrument at our disposal that allows us to identify tumor cells, especially when it comes to questions involving cell networks involving different cell types. Clarifying the cell type is particularly interesting in research on 3D cell cultures of the OC when interactions between different cell types or drug effects in the microenvironment are to be researched [26]. Influencing immune cells or fibroblasts are likely to differ in their diploid character from the OC cells, whose chromoanagenesis can be detected by the OC-FISH test. In imaging, immunofluorescence detection of expressed proteins is possible in combination with FISH signals [27,28]. The differentiation of tumor cells with cells from the surrounding area can also be validated in formalin-fixed paraffin-embedded (FFPE) thin-layer preparations using the OC-FISH test, thus providing greater analytic certainty [29].

## Figures and Tables

**Figure 1 biomedicines-12-01171-f001:**
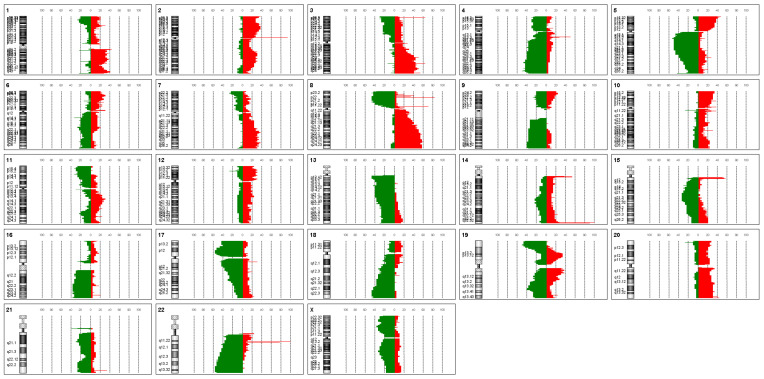
Shown is the relative proportion of copy number variations in 30 ovarian carcinomas for each human chromosome separately. Red bars represent gains, and green bars represent losses. The ideogram of the corresponding chromosome is shown on the left. The dashed lines show the percentage from 0 to 100% in increments of 10.

**Figure 2 biomedicines-12-01171-f002:**
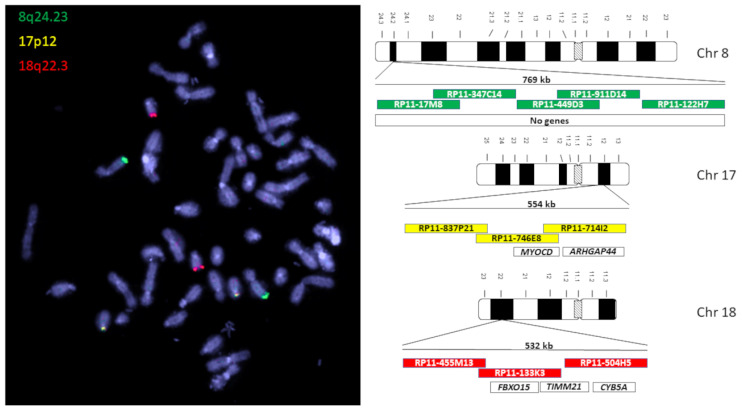
On the left is a lymphatic, diploid metaphase with the signals of the OC-FISH test. On the right are ideograms of the chromosomes of the OC-FISH test. The BACs that are involved in the respective loci are arranged in relation to each other in the colored boxes. In the uncolored boxes, genes are arranged in the respective OC-FISH test loci.

**Figure 3 biomedicines-12-01171-f003:**
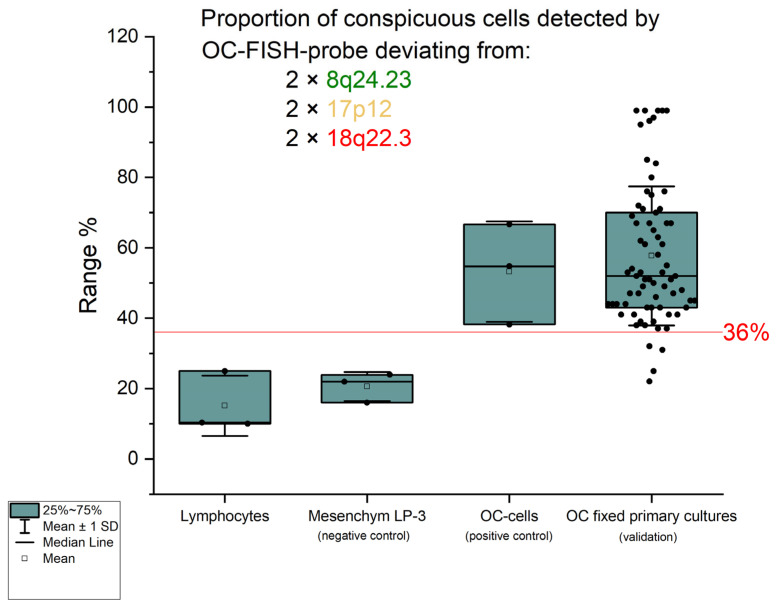
The graph shows the percentage of those interphases that deviate from the diploid pat-tern in the signal pattern. The green boxes correspond to the confidence interval (25%; 75%), the black solid line in the box corresponds to the median, and the transparent square in the box corresponds to the mean. The red solid line represents the cut-off of 36%. The standard deviation from the mean is shown by the antennas. The black dots are the percentages of interphases that are not diploid.

**Figure 4 biomedicines-12-01171-f004:**
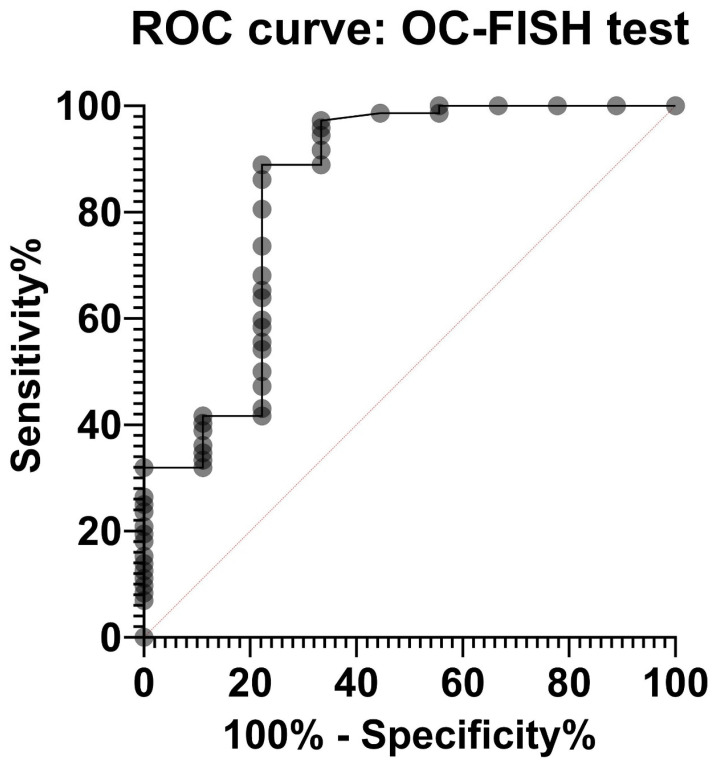
Receiver operating characteristic (ROC) curve of the OC-FISH test. The area under the curve is 0.8434 (95%CI: 0.672; *p* = 0.0008).

**Table 1 biomedicines-12-01171-t001:** All cells that were used as negative controls (green) and positive controls (red) and for validation (yellow) of the OC-FISH test are listed. The percentage of abnormal cells, the histological status, the FIGO value, the differentiation, and the score of the OC-FISH test are compared.

OC-FISH Test on:	Conspicuous Signal Pattern (%)	Histology	FIGO	Grading	Test Result
male lymphocytes	10				true negative
male lymphocytes	10				true negative
male lymphocytes	25				true negative
LP-3 mesenchyme	16				true negative
LP-3 mesenchyme	24				true negative
LP-3 mesenchyme	22				true negative
OvCa-cells	38	n.a.	n.a.	n.a.	true positive
OvCa-cells	67	n.a.	n.a.	n.a.	true positive
OvCa-cells	55	n.a.	n.a.	n.a.	true positive
OVCAR 3	100	serous	n.a.	3	true positive
OVCAR 8	100	serous	n.a.	3	true positive
SCOV 3	100	serous	n.a.	n.a.	true positive
RU-0001-OC	43	n.a.	n.a.	n.a.	true positive
SU-0002-OC	25	serous	IIIc	3	false negative
KA-0003-OC	65	serous	IIIb	borderline	true positive
WR-0004-OC	22	serous papillary	IIIc	3	false negative
BR-0005-OC	43	papillary	IIb	3	true positive
BI-0006-OC	39	serous	IIc	3	true positive
KS-0007-OC	32	serous	IIc	borderline	false negative
JH-0008-OC	50	serous papillary	IV	n.a.	true positive
BI-0009-OC	46	serous	IIc	3	true positive
HK-0011-OC	51	endometrioid	IIc	2	true positive
JE-0012-OC	49	endometrioid	IIIc	3	true positive
RH-0013-OC	58	papillary	IIIc	2	true positive
MH-0014-OC	43	serous papillary	n.a.	n.a.	true positive
SI-0015-OC	97	papillary clear cell	IIIc	3	true positive
DG-0016-OC	51	serous	IV	3	true positive
MA-0018-OC	37	serous papillary	n.a.	3	true positive
HJ-0019-OC	47	papillary	IIIc	3	true positive
KA-0020-OC	80	n.a.	IIIc	3	true positive
TW-0021-OC	53	serous papillary	IIIc	3	true positive
JD-0022-OC	85	serous papillary	IIIc	3	true positive
JS-0023-OC	38	serous papillary	IV	n.a.	true positive
KD-0024-OC	31	papillary	IIIc	2	false negative
SM-0028-OC	49	serous papillary	IIIc	3	true positive
JK-0030-OC	41	serous papillary	IIIc	2	true positive
ED-0031-OC	41	serous papillary	IIIb	3	true positive
SB-0032-OC	39	serous papillary	IIIb	3	true positive
TM-0033-OC	37	solid	IIIc	3	true positive
KA-0034-OC	53	serous papillary	IIIc	3	true positive
SD-0035-OC	75	mucinous	n.a.	2	true positive
KH-0036-OC	84	serous	n.a.	3	true positive
FF-0037-OC	67	papillary	IV	2	true positive
SD-0038-OC	55	mucinous	n.a.	2	true positive
PR-0039-OC	47	n.a.	Ic	2	true positive
RR-0040-OC	63	serous papillary	Iic	3	true positive
KR-0041-OC	61	papillary	IV	3	true positive
PK-0042-OC	47	serous	IV	3	true positive
DU-0043-OC	76	serous	n.a.	3	true positive
EA-0044-OC	96	serous papillary	IV	3	true positive
MI-0045-OC	99	n.a.	IV	n.a.	true positive
BE-0046-OC	52	n.a.	n.a.	n.a.	true positive
TB-0051-OC	70	serous papillary	IV	n.a.	true positive
SM-0053-OC	44	serous papillary	IV	n.a.	true positive
KK-0054-OC	99	endometrioid	Ia	1	true positive
BM-0055-OC	38	n.a.	n.a.	n.a.	true positive
AK-0056-OC	61	serous papillary	IV	1	true positive
MK-0057-OC	71	papillary	IIIc	3	true positive
TG-0058-OC	99	serous papillary	IIIc	3	true positive
JL-0059-OC	41	solid	IIIc	3	true positive
ZH-0061-OC	41	serous papillary	IV	n.a.	true positive
BF-0062-OC	48	papillary	IIIc	3	true positive
HH-0063-OC	51	serous papillary	IIc	3	true positive
SK-0064-OC	44	serous papillary	IIb	3	true positive
NG-0066-OC	43	serous papillary	n.a.	2	true positive
WP-0067-OC	62	serous	n.a.	borderline	true positive
PH-0068-OC	54	serous papillary	IV	3	true positive
BR-0069-OC	52	serous	IV	3	true positive
HM-0070-OC	95	endometrioid	Ic	2	true positive
SA-0071-OC	44	serous	IIIc	1	true positive
DR-0072-OC	45	mucinous	IV	1	true positive
KT-0073-OC	53	serous	IIc	borderline	true positive
AI-0075-OC	67	papillary	IV	1	true positive
KA-0076-OC	44	serous papillary	Ic	borderline	true positive
BKM-0077-OC	67	serous papillary	IV	3	true positive
WM-0078-OC	45	clear cell	n.a.	2	true positive
FG-0079-OC	71	serous papillary	IV	2	true positive
EL-0080-OC	99	n.a.	n.a.	n.a.	true positive
SB-0083-OiCa	72	serous papillary	IIIb	3	true positive
HK-0084-OC	67	serous endometrioid clear cell	IIIa	n.a.	true positive
BT-0085-OC	76	n.a.	n.a.	n.a.	true positive
HM-0086-OC	69	serous papillary	IIIc	2	true positive
ZI-0087-OC	99	serous papillary	IIIc	2	true positive

n.a. = not analyzed.

## Data Availability

The data presented in this study are available on request from the corresponding author.

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
