# Peer review of "Generation of a Specific Fluorescence In Situ Hybridization Test for the Detection of Ovarian Carcinoma Cells"

_biomedicines, 2024, doi:10.3390/biomedicines12061171_

Round 1

Reviewer 1 Report

Comments and Suggestions for Authors

The Manuscript by A. Limburg, X. Qian, B. Brechtefeld, N. Hedemann, I. Flörkemeier, C. Rogmans, L. Oliveira-Ferrer, N. Maass, N. Arnold, D.O. Bauerschlag, J.P. Weimer “Generation of a specific fluorescence in situ hybridization test for the detection of ovarian carcinoma cells” describes an elegant modern approach to visualization and identification of ovarian cancer cells. In recent years, fluorescence in situ hybridization technique have been finding a broad application in detection of chromosomal rearrangements. Thus, determination of specific chromosomal features of cancer cells and subsequent analysis by means of FISH represents a promising tool to identification of cancer and contributes to development of diagnostics and personalized medicine. The manuscript is written in a good scientific language, results are presented clearly.

Authors considered the reviewers’ questions to the previous version of the manuscript and made appropriate corrections in the text improving the scientific soundness of the article. The list of references should be corrected: the numbers of articles (refs. 9, 17, 21) and pages (ref. 20).

The article by A. Limburg and co-authors in its present form is suitable to publication Biomedicines and will be interesting for the broad auditory of the journal.

Author Response

We have checked the references given and corrected the spelling of some authors. We have corrected the page number for reference number 20.

Reviewer 2 Report

Comments and Suggestions for Authors

Generation of a specific fluorescence in situ hybridization test for the detection of ovarian carcinoma cells

 By Amelie Limburg and colleagues Is an interesting and well documented report. Genomic imbalances of 30 ovarian cancers in three genomic loci is detected and verified in 71 primary cultures of ovarian carcinoma. in the supplementary table there are also several cell lines known as OVCAR 3 and 8 and also SCOV (possibly as positive control). The fact that the validation of the results is also based on the genomic analysis of these cells, in table S1 all the details are noted. The results provided in these 2 additional tables S1 and S2 are crucial and better to be merged and brought into a single table among the text. In conclusion, this work can be accepted with minor modification.

Author Response

We combined the contents of the supplementary tables S1 and S2 into one table and inserted it into the article as a regular table.